# Nanocrystalline Suspensions of Irbesartan Enhance Oral Bioavailability by Improving Drug Solubility and Leading Endocytosis Uptake into the Intestine

**DOI:** 10.3390/pharmaceutics13091404

**Published:** 2021-09-03

**Authors:** Saori Deguchi, Fumihiko Ogata, Masaki Watanabe, Hiroko Otake, Naoki Yamamoto, Naohito Kawasaki, Noriaki Nagai

**Affiliations:** 1Faculty of Pharmacy, Kindai University, Higashi-Osaka 577-8502, Japan; 2045110002h@kindai.ac.jp (S.D.); ogata@phar.kindai.ac.jp (F.O.); 1611610018u@kindai.ac.jp (M.W.); hotake@phar.kindai.ac.jp (H.O.); kawasaki@phar.kindai.ac.jp (N.K.); 2Center for Clinical Trial and Research Support, Research Promotion and Support Headquarters, Fujita Health University, Toyoake 470-1192, Japan; naokiy@fujita-hu.ac.jp

**Keywords:** nanocrystalline, intestinal absorption, irbesartan, solubility, endocytosis

## Abstract

We attempted to design irbesartan nanocrystalline (IRB-NC) suspensions by the bead mill method, and we evaluated the bioavailability (BA) in the oral administration of the nanocrystalline drug. The mean particle size of the IRB-NC suspensions was approximately 140 nm, and the crystalline structure of irbesartan in these suspensions was different using the bead mill method. The aggregation and degradation of irbesartan were not observed for one month, and the solubility increased. Moreover, the inclusion complex formation of IRB-NC suspensions with 2-hydroxypropyl-β-cyclodextrin was higher than that in traditional IRB powder (IRB-P). In addition, the intestinal absorption of IRB-NC suspensions was higher than that of IRB-P suspensions, and the reducing effect on blood pressure in spontaneously hypertensive SHR-SP rats orally administered IRB-NC suspensions was significantly higher than in those administered IRB-P suspensions. On the other hand, the intestinal penetration of IRB-NC suspensions was attenuated by the inhibitors of clathrin-dependent endocytosis (CME). In conclusion, we improved the low oral BA of irbesartan by preparing IRB-NC suspensions and showed that both the solubility and CME are related to the enhanced intestinal absorption of IRB-NC suspensions, resulting in an increase in their antihypertensive effect. These findings provide significant information for the development of oral nanomedicines.

## 1. Introduction

The treatment of hypertensive subjects decreases the risk factors of cardiovascular diseases, since hypertension is related to the onset of cardiovascular diseases, such as stroke and coronary artery disease, and leads to an increase in mortality [1,2]. In clinical practice, nonselective beta blockers, thiazide diuretics, and angiotensin II type 1 (AT1) receptor blockers (ARBs) have been used as a form of therapy for hypertension, and irbesartan ARBs are widely employed in clinical settings.

The molecular weight of irbesartan is 428.53 g/mol, and the two pKa are 4.12 and 7.4 [3]; its small molecular active ingredient is a noncompetitive, potent, long-acting ARB and is specific to the AT1 receptor subtype [4,5,6,7]. In addition, it is shown in the package insert that the oral administration of irbesartan improves hypertension in stroke-prone spontaneously hypertensive rats (SHR-SP rats) and human patients with hypertension. On the other hand, irbesartan shows low aqueous solubility and high permeability, and is classed as a BCS class II drug [3,4,8]. Therefore, the intestinal absorption of irbesartan limits its dissolution rate, and its oral bioavailability (BA) is approximately 60–80% in patients [9]. Moreover, a high variation of oral BA in irbesartan has been observed among individuals [10]. For these reasons, improvement in the dissolution rates and oral BA of poorly soluble drugs is required.

To improve the oral BA in poorly soluble drugs, various strategies—including silica- and biodegradable nanoparticles, amorphous solids, micelles, cosolvents, salts, liposomes, dendrimers, cyclodextrin, polymer–drug conjugates, and nanocrystalline (NC) suspensions—have been utilized [11,12]. In these strategies, the use of NC suspensions is increased to improve the oral BA of a poorly soluble drug [13,14,15,16,17,18]. The methods for preparing NC suspensions are generally divided into the build-up and breakdown methods [19]. In the build-up method, a poorly soluble drug is dissolved using an organic solvent. Then, solvent evaporation or the addition of an antisolvent is performed, and the poorly soluble drug subsequently precipitates as nanoparticles. However, the build-up method has the following problems: low drug loading, a scaling-up challenge, and uncontrolled solid-state formation. Therefore, the bottom-up approach is not usually preferred [20]. In contrast, wet media milling and a high-pressure homogenizer are applied to decrease the particle size in the breakdown method, with a poorly soluble drug suspended in an aqueous medium or a combination of aqueous and nonaqueous media. Previously, we also prepared an oral formulation based on NC suspensions and showed that the transintestinal penetration of indomethacin nanoparticles is higher in comparison to that of suspensions containing indomethacin microparticles and an indomethacin solution (dissolved indomethacin) [21,22]. In addition, we reported that energy-dependent endocytosis is related to the trafficking pathways of drug nanoparticles [21,23].

We previously prepared nanocrystal dispersions using similar treatments and additives [21,22,23]. However, even with the same treatment, if the drugs are different, the resulting particle sizes and absorption mechanisms are different. Furthermore, the crystalline structures are also different. Therefore, it is important to clarify the physical characteristics, absorption, and drug efficacy of nanocrystals using drugs with different properties for designing and applying nanocrystal dispersions. Based on this, this study—using irbesartan ARB, which is classed as a BCS class II drug—will lead the way for further studies aiming to develop oral nanomedicines. In this study, we attempted to design irbesartan nanocrystalline (IRB-NC) suspensions using the bead mill method, and we evaluated the intestinal absorption of IRB-NC suspensions. In addition, we demonstrated the usefulness of antihypertensive drugs in IRB-NC suspensions using a stroke-prone spontaneously hypertensive (SHR-SP) rat model [24,25].

## 2. Materials and Methods

### 2.1. Animals

Six-week-old male Wistar rats (approximately 200 g) were obtained from Kiwa Laboratory Animals Co., Ltd. (Wakayama, Japan), and 14-week-old male SHR-SP rats (approximately 340 g) were purchased from the Faculty of Medicine of Kindai University. They were housed under controlled temperature (25 °C) and lighting (7:00–19:00 light; 19:00–7:00 dark) conditions, with unlimited access to a standard CE-2 formulation diet (Clea Japan Inc., Tokyo, Japan) and drinking water. The guiding principles were approved by the Japanese Pharmacological Society, and all experiments were performed in accordance with the guidelines for animal experimentation of the international association for the study. The experiments were approved on 1 April 2019 (project identification code, KAPS-31-014) by Kindai University.

### 2.2. Chemicals

Irbesartan powder (IRB-P), propyl p-hydroxybenzoate, and cytochalasin D were purchased from Wako Pure Chemical Industries, Ltd. (Osaka, Japan). Rottlerin and dynasore were provided from Nacalai Tesque (Kyoto, Japan), and nystatin was obtained from Sigma-Aldrich Japan (Tokyo, Japan). A Bio-Rad Protein Assay Kit was purchased from BIO-RAD (Hercules, CA, USA). Pentobarbital and methylcellulose (MC) were obtained from Tokyo Chemical Industry Co., Ltd. (Tokyo, Japan), and Shin-Etsu Chemical Co., Ltd. (Tokyo, Japan), respectively. 2-Hydroxypropyl-β-cyclodextrin (HPβCD) was provided by Nihon Shokuhin Kako Co., Ltd. (Tokyo, Japan). All other chemicals were of the highest purity commercially available.

### 2.3. Production of IRB-NC Suspensions

Suspensions containing IRB-P (IRB-P suspensions) were prepared by mixing irbesartan powder, MC, and HPβCD in purified water, produced following our previous report [22,26]. Briefly, IRB-P and MC were milled by an agate mortar for 30 min; subsequently, the mixture was transferred into a 1.5 mL tube containing 1 mm zirconia beads and crushed using Bead Smash 12 (Wakenyaku Co. Ltd., Kyoto, Japan) at 3000 rpm for 30 s at 4 °C. Then, a HPβCD solution was added to the 1.5 mL tube containing the mixture, which was stirred for 4 h and milled by 0.1 mm zirconia beads via Bead Smash 12 at 5500 rpm for 30 s × 30 times at 4 °C (IRB-NC suspensions). In the stirring and bead milling treatment, adsorption and inclusion complex formation of irbesartan with HPβCD was performed. The compositions of the IRB-P and IRB-NC suspensions were as follows: 0.2% irbesartan, 0.5% MC, and 5% HPβCD in purified water.

### 2.4. Measurement of Irbesartan by the HPLC Method

An LC-20AT system (HPLC, Shimadzu Corp., Kyoto, Japan) equipped with a 2.1 × 50 mm Inertsil^®^ ODS-3 column (GL Science Co., Inc., Tokyo, Japan) was used to measure the irbesartan concentration. Acetonitrile/0.1% formic acid (37/63, *v*/*v*%) was applied as a mobile phase, and flowed at 0.25 mL/min. Moreover, the wavelength for detection and an internal standard were selected as 254 nm and 1 µg/mL propyl p-hydroxybenzoate, respectively. The measurement was performed at 35 °C using a column oven, and the samples (10 μL) were injected by SIL-20AC.

### 2.5. Evaluation of the Characteristics in the IRB-NC Suspensions

The characteristics—such as particle size, form, and number, crystalline structure, melting point, zeta potential, solubility, and dispersibility—were analyzed following our previous study [21,22,23,26]. A laser diffraction particle size analyzer, SALD-7100 (Shimadzu Corp., Kyoto, Japan), and the Dynamic Light Scattering NANOSIGHT LM10 (QuantumDesign Japan, Tokyo, Japan) were used to determine the particle size distribution. The particle form was observed by SPM-9700 (Shimadzu Corp., Kyoto, Japan), and the provided phase and height images of irbesartan were combined and expressed as atomic force microscopic (AFM) images. Moreover, the particle number was measured by NANOSIGHT LM10. The crystalline form of the lyophilized IRB-P and IRB-NC suspensions was analyzed by a powder X-ray diffraction (XRD) method using Mini Flex II (Rigaku Co., Tokyo, Japan), and the melting point was evaluated by thermogravimetry–differential thermal analysis (TG-DTA) measurements under a nitrogen atmosphere using the simultaneous TG-DTA apparatus DTG-60H (Shimadzu Corp., Kyoto, Japan). The zeta potential was measured by a micro-electrophoresis zeta potential analyzer, model 502 (Nihon Rufuto Co., Ltd., Tokyo, Japan). In addition, the solubility of irbesartan in the IRB-NC suspensions was measured as follows: the IRB-NC suspensions were centrifuged at 100,000× *g* using a Beckman Optima^TM^ MAX-XP Ultracentrifuge (Beckman coulter, Osaka, Japan) to separate into soluble and nonsolubilized irbesartan. After this, the concentration of soluble irbesartan was measured by the HPLC method described above. In this study, the concentration of soluble irbesartan is expressed as the solubility of irbesartan. Then, 3 mL of the IRB-NC suspensions in a 5 mL test tube were incubated in the dark at 22 °C for one month (30 days) to evaluate the dispersibility, which was determined by measurement of the irbesartan concentration in the sample collected from the upper 90% of the test tube over time.

### 2.6. In Vitro Intestinal Penetration of Irbesartan in Rats

Seven-week-old male Wistar rats (approximately 220 g) were euthanized by injection of a lethal dose of pentobarbital, and the small intestines were washed in saline and collected. The removed small intestines (the jejunum part) were set on a methacrylate cell, and the apical and basolateral sides were filled with IRB-NC suspensions and a pH 7.4 buffer consisting of 10 mM HEPES, 136.2 mM NaCl, 1 mM K_2_HPO_4_, 1.7 mM CaCl_2_, 5.5 mM glucose, and 5.3 mM KCl. The experiments were performed at 4 °C, the condition under which energy-dependent endocytosis was inhibited [27], or at 37 °C (normal condition). Moreover, four pharmacological inhibitors (54 µM nystatin (CavME inhibitor) [28], 40 µM dynasore (CME inhibitor) [29], 2 µM rottlerin (MP inhibitor) [30], or 10 µM cytochalasin D (phagocytosis inhibitor) [28]) were used to clarify the relationships between the endocytosis pathways. The pharmacological inhibitors in 0.5% DMSO were pretreated for 5 min, and then treated by the addition of a pH 7.4 buffer in a basolateral chamber. In this study, 0.5% DMSO was used as a vehicle, and the irbesartan concentration in the sample was determined by the HPLC method described above. Moreover, the area under the drug concentration–time curve in the basolateral chamber (*AUC*_0–6h_) was analyzed according to the trapezoidal rule. In general, in an in vitro intestinal penetration study, using small intestines, the membrane damage causes a sharp increase in the drug permeation (burst). Therefore, in this study, the experiment was performed in 6 h without a rapid increase (burst).

### 2.7. Measurement of the Irbesartan Concentration in Rat Blood

On the day before administration of the irbesartan suspensions, a cannula filled with 30 μg/mL of heparin (silicone tubing; i.d. 0.5 mm, o.d. 1.0 mm) was inserted into the right jugular vein of the rats under isoflurane anesthesia. The suspensions containing irbesartan (0.2 mg/kg) were orally administered to 7-week-old male Wistar rats fasted for 8 h (approximately 220 g), and 200 μL of venous blood was collected at 0, 0.5, 1, 2, 6, and 24 h from the jugular vein through the cannula after the oral administration of irbesartan. The provided blood was centrifuged at 800× *g* for 15 min at 4 °C, and the plasma irbesartan was measured by the HPLC method described above. The area under the blood concentration–time curve (*AUC*_0–24h_) was analyzed according to the trapezoidal rule.

### 2.8. Measurement of Blood Pressure (BP) in SHR-SP Rats Orally Administered Irbesartan

The suspensions containing irbesartan (0.2 mg/kg) were orally administered to 14-week-old male SHR-SP rats (approximately 340 g), and systolic blood pressure (SBP), diastolic blood pressure (DPB), and heart rate were measured at 0, 1, 2, 6, and 24 h after the oral administration of irbesartan using a noninvasive blood pressure analysis system, BP-98A (Softron, Tokyo, Japan). Moreover, the difference in BP (SBP and DBP) in the rats administered suspensions either containing irbesartan or not are expressed as ΔSBP and ΔDBP, respectively, and the area under the ΔBP (ΔSBP and ΔDBP)–time curve (*AUC*_ΔSBP_ and *AUC*_ΔDBP_) for 0–24 h was analyzed according to the trapezoidal rule. In this study, the SBP and DPB in SHR-SP rats were 261 ± 10.8 mmHg and 148 ± 6.5 mmHg, respectively.

### 2.9. Statistical Analysis

The data are expressed as the mean ± standard error of the mean (S.E.M), and the experimental numbers (n) are shown in the figure legends. A statistical analysis was performed using Student’s t-test (two groups) and one-way analysis of variance (ANOVA) followed by Dunnett’s multiple comparison (multiple groups). Moreover, two-way ANOVA was applied in the statistical analysis of particle size frequencies and solubility of irbesartan in the IRB-NC suspensions. A *p*-value of < 0.05 was chosen as the significance level.

## 3. Results

### 3.1. Changes in the Characteristics of Irbesartan Particles with or without Bead Mill Treatment

We attempted to produce IRB-NC suspensions by bead mill treatment, and we then evaluated the characteristics of these IRB-NC suspensions. Figure 1A–D shows the particle size frequencies of irbesartan with or without bead mill treatment. The mean particle size of irbesartan in the IRB-P suspensions was 4.89 ± 0.49 µm (Figure 1A), which was decreased to approximately 140 nm by bead mill treatment (Figure 1B,C), and the nanoparticles were sphere-shaped (Figure 1D). Furthermore, we investigated the effect of bead mill treatment on drug solubility (Figure 1E). The drug solubility of irbesartan was increased to 0.306 µM from 0.188 µM by bead mill treatment. Moreover, the inclusion complex formation of irbesartan with HPβCD was also increased by the bead mill treatment. The drug solubility of the mixture containing IRB-P and HPβCD was 2.1-fold that of IRB-P without HPβCD. On the other hand, the drug solubility of the mixture containing the IRB-NC suspensions and HPβCD was 3.0-fold that without HPβCD. The ratio of dissolved irbesartan was only 0.02% in the 0.2% irbesartan suspensions. Next, the changes in the crystalline structure and stability of irbesartan with or without bead mill treatment were evaluated. Figure 2 shows the powder X-ray diffraction patterns and the TG-DTA curve of the IRB-P and IRB-NC suspensions. The crystalline form and melting point of irbesartan were changed by bead mill treatment, and the peak patterns of XRD were different between IRB-P and IRB-NC. The IRB-P and IRB-NC in the suspensions took on forms A and B, respectively. The melting point of the irbesartan reagent without additives and bead mill treatment was 182 °C, and the addition of additives (MC and HPβCD) changed the melting point to 177 °C from 182 °C. On the other hand, the melting point was similar between the IRB-P and IRB-NC suspensions, regardless of the differences in crystalline form. Figure 3 shows the stability of irbesartan in the IRB-NC suspensions one month after preparation. However, degradation of irbesartan was not observed in either the IRB-P or the IRB-NC suspensions; the irbesartan particles in the IRB-P suspensions were aggregated and precipitated one month after preparation. In contrast to the results of the IRB-P suspensions, the aggregation of irbesartan was not observed for one month in the IRB-NC suspensions. The zeta potential of the IRB-P or IRB-NC suspensions was −62.3 ± 4.9 mV and −57.8 ± 4.5 mV, respectively (*n* = 6).

### 3.2. Relationships between Energy-Dependent Endocytosis and the Transintestinal Penetration of the IRB-NC Suspensions in Rat Intestines

Our previous studies reported that drug nanoparticles are taken up into the tissue by energy-dependent endocytosis [21,23]. Therefore, we investigated whether energy-dependent endocytosis was related to the transintestinal penetration of IRB-NC suspensions (Figure 4). It was known that energy-dependent endocytosis is inhibited under cold temperatures [27]. In this study, the transintestinal penetration of the IRB-NC suspensions was significantly higher than that of IRB-P at 37 °C (*AUC*_0–6h_, 32.9 ± 6.7 nmol∙h/cm^2^), and the high penetration levels of the IRB-NC suspensions were strongly prevented at 4 °C. Moreover, the transintestinal penetration of the IRB-NC suspensions was attenuated by treatment with dynasore. In contrast, the pharmacological inhibitors nystatin, rottlerin, and cytochalasin D did not affect the transintestinal penetration of the IRB-NC suspensions. Furthermore, we measured the number of irbesartan solid particles (IRB-NC) in the basolateral chamber; however, no IRB-NC suspensions were detected at 4 and 37 °C.

### 3.3. Changes in the Therapeutic Effect of the IRB-NC Suspensions in the Intestines of SHR-SP Rats

It is important to clarify the differences in blood kinetics between the oral administration of the IRB-P and IRB-NC suspensions. Figure 5 shows the plasma irbesartan concentration in the rats orally administered the IRB-P and IRB-NC suspensions. The *C*_max_ of irbesartan was previously observed approximately 1.5–2 h after oral administration [9], and the behavior of the plasma irbesartan concentration in this study was similar to this previous study. Furthermore, the plasma irbesartan levels in those rats orally administered the IRB-NC suspensions were significantly higher than in those administered the IRB-P suspensions, and the *AUC*_0–24h_ in the IRB-NC suspensions was 1.84-fold higher than that in the IRB-P suspensions. We also demonstrated the effect of the IRB-NC suspensions on the BP in SHR-SP rats (Figure 6). Similar to the results of the absorption profile (Figure 5), the BP-reducing effect was enhanced by bead mill treatment, and the *AUC*_ΔSBP_ and *AUC*_ΔDBP_ in rats orally administered the IRB-NC suspensions were 1.53- and 1.33-fold higher than those administered the IRB-P suspensions, respectively.

## 4. Discussion

Using NC suspensions, which are submicron colloidal suspensions of poorly soluble drugs in media with or without a nonaqueous solution [31], enhances the dissolution rates and oral BA of poorly soluble drugs. In this study, we prepared IRB-NC suspensions by the bead mill method and found that the solubility in these IRB-NC suspensions was improved, and that these IRB-NC suspensions were taken up by the CME, resulting in an increase in transintestinal penetration. In addition, we showed that the oral administration of IRB-NC suspensions enhances the BP-reducing effect in comparison to traditional formulations (IRB-P suspensions) (Figure 7).

Our previous study showed that it is possible to prepare suspensions containing nanoparticles of poorly soluble drugs in a solution containing MC and HPβCD by bead mill treatment [22,26]. MC enhances the crushing efficiency in the bead mill, and HPβCD works to prevent the aggregation of drug particles [32]. According to these previous methods, we attempted to design IRB-NC suspensions. The particle size of the IRB-P in the suspensions was 1–55 µm (Figure 1A), and the irbesartan particles were maintained in the nanosize order (approximately 140 nm) by bead mill treatment (Figure 1B,C). Previous reports have shown that the drug solubility and the inclusion ratio of drug–HPβCD in the NC suspensions increases [22]. In this study, both the solubility and the inclusion ratio of drug–HPβCD in the IRB-NC suspensions were significantly higher than that in the IRB-P suspensions. Similarly to HPβCD, it is known that the crystalline structure is also related to solubility, and irbesartan has two crystalline structures, forms A and B, and the aqueous solubility of form B is lower than that of form A, i.e., 0.011 and 0.0006 mg/mL, respectively [7,8,33]. Therefore, we demonstrated whether the crystalline structure of irbesartan is changed by bead mill treatment. The peak patterns of XRD in the suspensions containing IRB-P were similar to those in the IRB reagent, and the crystalline form was form A (Figure 2A,B). However, the XRD patterns were different between the IRB-P and IRB-NC suspensions, and the crystalline form was changed to form B from form A by bead mill treatment (Figure 2B). We also measured the melting point of the IRB-P and IRB-NC suspensions (Figure 2C,D). It was discovered that the melting point of forms A and B in irbesartan is 182 and 186 °C, respectively [33]. The melting point of the irbesartan reagent without additives or bead mill treatment was 182 °C, and these data are the same as the previous data of Araya-Sibaja et al. [33]. However, the melting point of the IRB-P mixture containing MC and HPβCD (additives) decreased to 177 °C (Figure 2C). Moreover, the melting point of the IRB-NC suspensions containing additives was also different to the melting point of form B, and the melting points of both the IRB-P and the IRB-NC suspensions were similar (Figure 2D). The decrease in melting point may have been caused by the MC and HPβCD. From the point of view of solubility, form A of irbesartan is more effective in improving BA than form B. However, our results showed that the treatment using a bead mill and additives (MC and HPβCD) can be applied to suspensions containing approximately 60–250 nm of IRB-NC, with these IRB-NC suspensions providing high solubility in comparison to the IRB-P suspensions. From these results, it is suggested that controlling the particle size below 250 nm has a stronger effect on solubility than the regulation of the crystalline form in irbesartan, and these changes may enhance the dissolution rates and oral BA of poorly soluble drugs.

Drug stability is important to make use of DDS-system-based NC suspensions. Generally, nanoparticles have low degradability, but are easily aggregated. In contrast, it was discovered that polymeric stabilizers and small molecule surfactants lead to kinetic stability of NC suspensions, and our previous study also reported that combination with HPβCD prevents the aggregation of drug nanoparticles [32]. In this study, no degradation, precipitation, or aggregation was observed in the IRB-NC suspensions for one month after their preparation (Figure 3), and the zeta potential in the IRB-NC suspensions was similar to that in the IRB-P suspensions. These results support our previous studies, and the prescription proposed in this study can be prepared as an oral formulation containing 0.2% IRB-NC suspensions.

Next, we measured the oral BA of irbesartan in the IRB-NC suspensions. The oral BA was enhanced by bead mill treatment, and the *AUC*_0–24h_ levels in rats orally administered IRB-NC suspensions were 1.84-fold higher than those without bead mill treatment (Figure 5). Furthermore, we measured the number of irbesartan particles in the basolateral chamber, and no irbesartan particles were detected. Taken together, the results suggest that this enhancement of oral BA may be caused by an increase in solubility. Another factor related to the improvement in transintestinal penetration is that energy-dependent endocytosis enhances the transintestinal penetration of drug nanoparticles [21,23]. Therefore, we also investigated whether IRB-NC suspensions are endocytosed in the intestine using cold-temperature incubation (4 °C) [27] and four pharmacological inhibitors of energy-dependent endocytosis, namely, nystatin [28], dynasore [29], rottlerin [30], and cytochalasin D [28]. The transintestinal penetration of irbesartan in IRB-NC suspensions was strongly prevented under cold temperature incubation, and dynasore inhibited the transintestinal penetration of the IRB-NC suspensions (Figure 4). The particle size corresponding to the CME inhibited by dynasore was approximately 120 nm [34], and the mean particle size of the irbesartan in the suspensions was approximately 60–250 nm in the NANOSIGHT LM10 and AFM images (Figure 1C,D). These results suggest that the high oral BA of the irbesartan in the IRB-NC suspensions was caused by both the enhancement of solubility and the energy-dependent uptake via CME, and the IRB-NC suspensions were dissolved in the intestine and released into the blood, since no solid nanoparticles (IRB-NC) were detected in the basolateral side. On the contrary, the transintestinal penetration levels of the IRB-NC suspensions under the cold-temperature incubation were lower than those of the dynasore treatment group. Cold-temperature incubation inhibits all energy-dependent uptake and some energy-independent uptake [27]. Therefore, other pathways, such as energy-independent uptake, may be related to the transintestinal penetration of IRB-NC suspensions. In addition, rats generally have a high bile concentration in the small intestine, even in a fasted state, and the rapid dissolution of drugs from nanocrystals into bile micelles may also be related to high plasma irbesartan concentration in the rats administered IRB-NC. Further study is needed to clarify the details of the transintestinal penetration of IRB-NC suspensions.

Irbesartan is an angiotensin receptor blocker, used for treating cardiac arrhythmia, cardiac insufficiency, and hypertension [9,35]. Pouleur [35] reported, in a clinical study, that irbesartan shows a dose-related reduction in trough SBP and DBP, demonstrating a decrease in BP of 10 mmHg from baseline and an approximately 70% higher BP in patients treated with a 300 mg dose of irbesartan compared to nontreated patients [35]. We evaluated the therapeutic effect of irbesartan by measuring BP, and it is important to select the correct model for investigating the usefulness of antihypertensive drugs. Spontaneously hypertensive (SHR) rats, an inbred strain for genetic hypertension, and SHR-SP rats, a substrain of SHR rats, have been used in many studies. Hypertension in this strain of rats starts at eight weeks, and their BP was higher in comparison to normotensive Wistar rats. In addition, SHR-SP rats have a much higher BP level than SHR rats, with SBP-SP rats being able to reach 250 mmHg [24,25]. In this study, SHR-SP rats were used to evaluate the differences in the therapeutic effects of irbesartan in IRB-P and IRB-NC suspensions. Both the SBP and DBP were reduced by the oral administration of irbesartan, and the reducing effect in SHR-SP rats orally administered IRB-NC suspensions was significantly higher than that in IRB-P suspensions (Figure 6). This result is consistent with the results of the absorption profile (Figure 5). It was estimated that many reagents in the discovery phase would show low oral BA due to poor aqueous solubility and incomplete absorption, and it is difficult to find a place for them on the market due to these limitations [36]. However, NC suspensions using MC and HPβCD may improve these problems.

## 5. Conclusions

We improved the low oral BA of irbesartan by preparing IRB-NC suspensions, and we showed that both high solubility and energy-dependent endocytosis are related to the enhanced transintestinal penetration and BA of IRB-NC suspensions, resulting in an increase in the antihypertensive effect (Figure 7). The mechanism of transintestinal penetration via energy-dependent endocytosis is hypothesized to be IRB-NC suspensions being taken up into the intestinal epithelium by CME, and then captured IRB-NC shifting to the blood through the dissolve and diffuse functions in the intestine. This study, which designed IRB-NC suspensions, will lead the way for further studies aiming to develop oral nanomedicines.

## Figures and Tables

**Figure 1 pharmaceutics-13-01404-f001:**
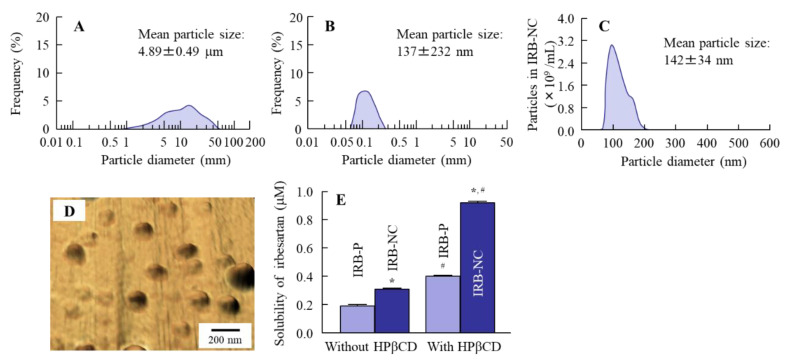
Particle size frequencies and solubility of irbesartan in the IRB-NC suspensions. (**A**) Particle size frequencies of irbesartan in the IRB-P suspensions. (**B**,**C**) Particle size frequencies of irbesartan in the IRB-NC suspensions. Data in (**A**,**B**) were measured using a laser diffraction particle size analyzer (SALD-7100), and those in (**C**) were obtained by the dynamic light scattering method (NANOSIGHT LM10). (**D**) AFM image of irbesartan in the IRB-NC suspensions. (**E**) Solubility of irbesartan in the IRB-NC suspensions with or without HPβCD. *n* = 6. * *p* < 0.05 vs. IRB-P for each category. ^#^ *p* < 0.05 vs. IRB without HPβCD for each category. The particle size of the IRB-NC suspensions was approximately 60–250 nm. The inclusion complex formation of the IRB-NC suspensions with HPβCD was higher than that of IRB-P, and the solubility increased.

**Figure 2 pharmaceutics-13-01404-f002:**
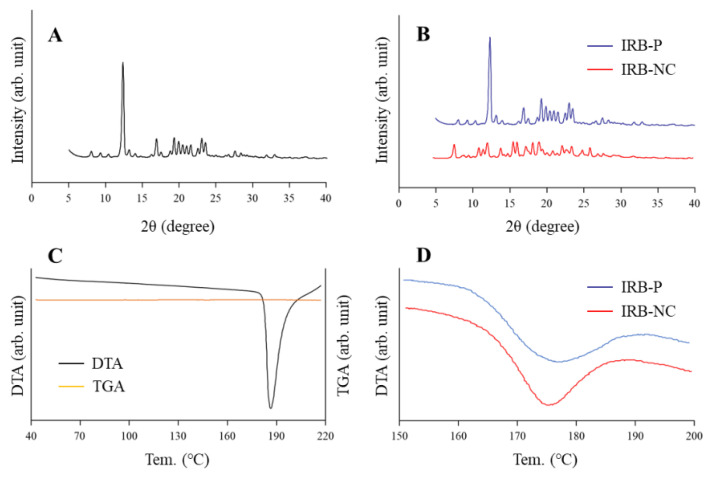
Analysis of the crystalline structure in irbesartan treated with or without a bead mill. (**A**) The powder X-ray diffraction patterns of the irbesartan reagent without additives and bead mill treatment. (**B**) The powder X-ray diffraction patterns of IRB-P and IRB-NC in the suspensions. (**C**) The TG-DTA curves of IRB-P without additives and bead mill treatment. (**D**) The DTA curves of the IRB-P and IRB-NC suspensions. The peak patterns of XRD were different between the IRB-P and IRB-NC suspensions. The melting point of IRB-P shifted to 177 °C from 182 °C upon the addition of additives, although no difference was observed between the IRB-P and IRB-NC suspensions.

**Figure 3 pharmaceutics-13-01404-f003:**
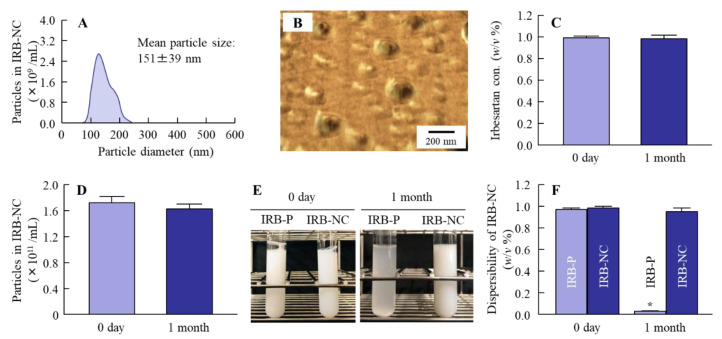
Stability of irbesartan in the IRB-NC suspensions one month after preparation. (**A**,**B**) Particle size frequencies (**A**) and AFM image (**B**) of irbesartan in the IRB-NC suspensions. (**C**,**D**) Changes in the concentration (**C**) and particle number (**D**) of irbesartan in the IRB-NC suspensions. (**E**,**F**) Representative pictures (**E**) and dispersibility (**F**) of the IRB-P and IRB-NC suspensions one month after preparation. *n* = 6. * *p* < 0.05 vs. 0 days for each category. The aggregation and degradation of irbesartan in the IRB-NC suspensions were not observed for one month.

**Figure 4 pharmaceutics-13-01404-f004:**
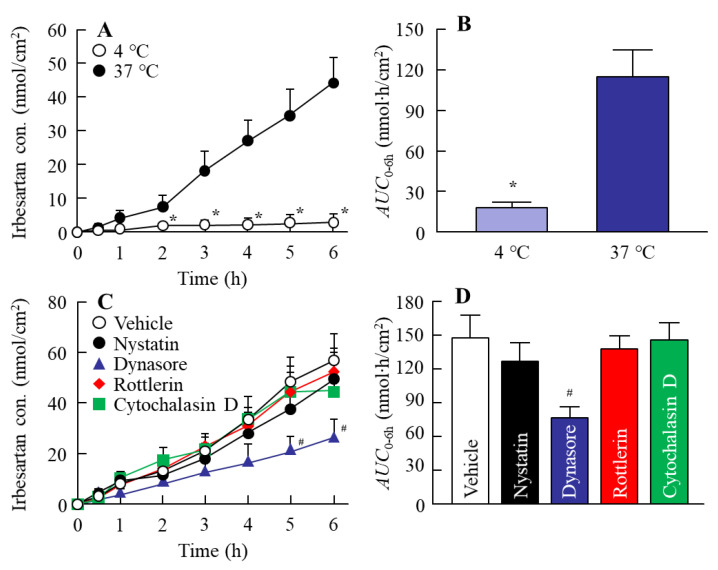
Effect of endocytosis inhibitor on the in vitro intestinal penetration of the IRB-NC suspensions. (**A**,**B**) Penetration profile (**A**) and *AUC*_0–6h_ (**B**) of irbesartan in the IRB-NC suspensions under 4 or 37 °C conditions. (**C**,**D**) Drug penetration profile (**C**) and *AUC*_0–6h_ (**D**) of the IRB-NC suspensions in the intestines of rats treated with endocytosis inhibitors. 4 °C, IRB-NC-suspension-treated intestine at 4 °C. 37 °C, IRB-NC-suspension-treated intestine at 37 °C. Vehicle, intestine cotreated with IRB-NC suspensions and a vehicle of endocytosis inhibitors (0.5% DMSO). Nystatin, intestine cotreated with IRB-NC suspensions and nystatin. Dynasore, intestine cotreated with IRB-NC suspensions and dynasore. Rottlerin, intestine cotreated with IRB-NC suspensions and rottlerin. Cytochalasin D, intestine cotreated with IRB-NC suspensions and cytochalasin D. *n* = 5–9. * *p* < 0.05 vs. 37 °C group for each category. ^#^
*p* < 0.05 vs. vehicle for each category. The intestinal penetration of the IRB-NC suspensions was attenuated by treatment with dynasore.

**Figure 5 pharmaceutics-13-01404-f005:**
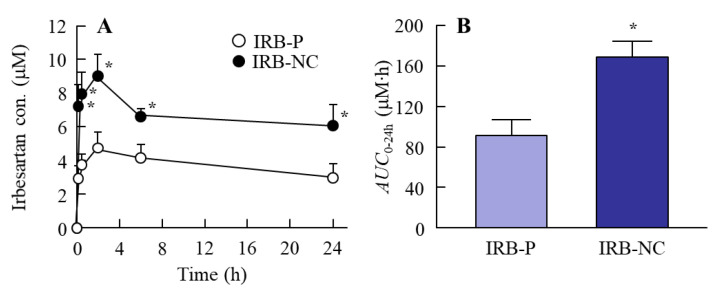
Changes in the plasma irbesartan concentration in the rats orally administered the IRB-P and IRB-NC suspensions. (**A**,**B**) The absorption profile (**A**) and *AUC*_0–24h_ (**B**) of irbesartan in the IRB-P and IRB-NC suspensions. IRB-P, IRB-P-suspension-administered rats. IRB-NC, IRB-NC-suspension-administered rats. *n* = 7. * *p* < 0.05 vs. IRB-P for each category. The intestinal absorption of irbesartan in the IRB-NC suspensions was significantly higher than that in the IRB-P suspensions.

**Figure 6 pharmaceutics-13-01404-f006:**
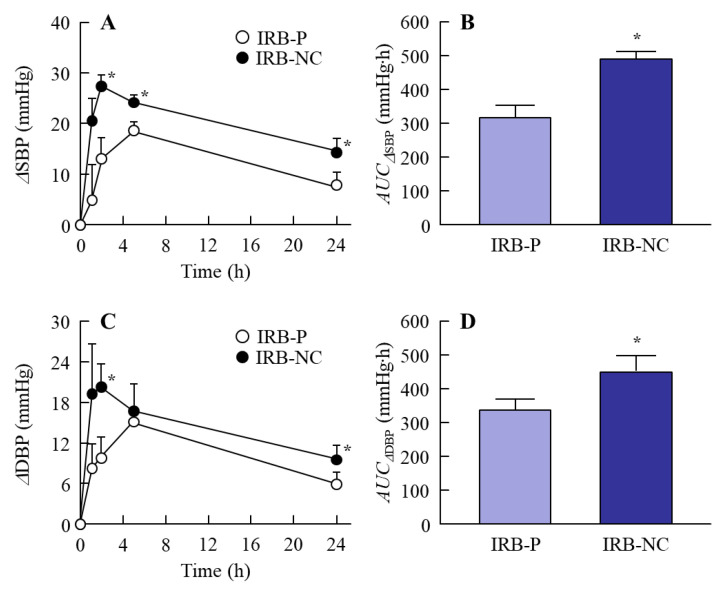
Changes in BP in the rats orally administered the IRB-NC suspensions. (**A**,**B**) ΔSBP profile (**A**) and *AUC*_Δ__SBP_ (**B**) in the rats orally administered the IRB-P and IRB-NC suspensions. (**C**,**D**) ΔDBP profile (**C**) and *AUC*_Δ__DBP_ (**D**) in the rats orally administered the IRB-P and IRB-NC suspensions. IRB-P, IRB-P-suspension-administered SHR-SP rats. IRB-NC, IRB-NC-suspension-administered SHR-SP rats. *n* = 7. * *p* < 0.05 vs. IRB-P for each category. The BP-reducing effect of the IRB-NC suspensions was significantly higher than that of the IRB-P suspensions.

**Figure 7 pharmaceutics-13-01404-f007:**
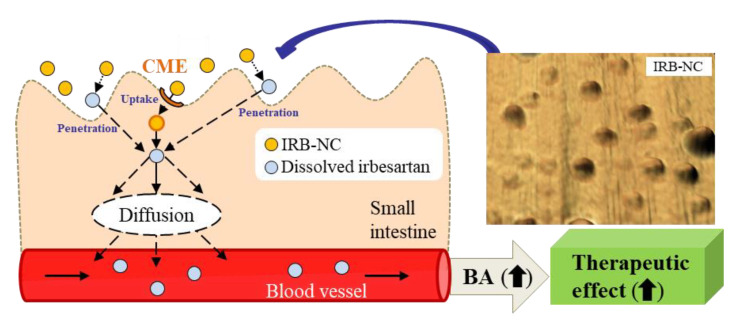
Mechanism of the intestinal penetration pathway and the hypertension preventive effect following the oral administration of IRB-NC suspensions.

## Data Availability

Not applicable.

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
