# Peer review of "Nanocrystalline Suspensions of Irbesartan Enhance Oral Bioavailability by Improving Drug Solubility and Leading Endocytosis Uptake into the Intestine"

_pharmaceutics, 2021, doi:10.3390/pharmaceutics13091404_

Round 1
Reviewer 1 Report
Although the topic of this manuscript is interesting, various amendments are required before it can accepted.
1) English writing requires polishing.
2) How did the authors prepare the HPbetaCD inclusion complex? The details need to be provided.
3) How did the authors assess the solubility? The details need to be provided.
4) Why one way ANOVA is applied? Fig 1E should apply two way ANOVA.
5) The cannulation information is missing.
Author Response
We carefully revised our manuscript according to the suggestions of the reviewer 1, and details are as follows.
< Q and A for Reviewer 1>
Q1. English writing requires polishing.
A1. Thank you very much for pointing this out. In order to respond to the reviewer’s comment, the manuscript was checked and edited by a native English-speaking person with sufficient scientific knowledge via MDPI English editing service (ID. English-33765).
Q2. How did the authors prepare the HPbetaCD inclusion complex? The details need to be provided.
A2. The reviewer’s comments are very important. The inclusion complex formation of irbesartan with HPβCD was performed by the stirring and bead milling treatment. We added the protocol in the Materials and Methods (line 117-120).
Q3. How did the authors assess the solubility? The details need to be provided.
A3. The reviewer’s comment is correct. The IRB-NC suspensions were centrifuged at 100,000 g using a Beckman OptimaTM MAX-XP Ultracentrifuge (Beckman coulter, Osaka, Japan), and separated to the soluble and nonsolubilized irbesartan. After that, the concentration of soluble irbesartan was measured by the HPLC method. In this study, the concentration of soluble irbesartan is expressed as solubility of irbesartan. In order to respond to the reviewer’s comment, we added the detail protocol in the Materials and Methods (line 151-154).
Q4. Why one way ANOVA is applied? Fig 1E should apply two way ANOVA.
A4. The reviewer’s comment is correct. We applied two-way ANOVA in the Figure 1E. Thank you very much for pointing this out. (line 207-210).
Q5. The cannulation information is missing.
A5. Thank you for pointing out this. On the day before administration of irbesartan suspensions, a cannula filled with 30 μg/mL heparin (silicone tubing; i.d. 0.5 mm, o.d. 1.0 mm) was inserted into the right jugular vein of rats under isoflurane anesthesia. In order to respond to the reviewer’s comment, we added the protocol in the Materials and Methods (line 181-183).
Thank you for great comments.

Reviewer 2 Report
- The authors should discuss a bit more detail about novelty of the work, as it seems that they have used a similar concept that was obtained in their previous works and that drug compound was just changed to IRB.
- In vitro penetration study. Did you also evaluate TEER in that chamber study? If not, a possibility to change the paracellular permeability in these treatments should be discussed in the manuscript.
- Rat pharmacokinetic and pharmacodynamic studies. As the authors know, prandial states (fasted or fed) generally have a big impact on solubility of poorly soluble drugs in the gut, so these information should be clarified in Method.
- Figure 4. A comparison between IRB-NC and IRB-P under 37 deg-C should also be performed.
- Figure 5. Rats generally have a high bile concentration in the small intestine even in fasted state. I guess IRB has a high log P value and would be easily incorporated into the bile micelles. Although according to the the in vitro study CME have an important role in penetration of the drug in the intestinal epithelium, do the authors think that the CME also is a major factor in in vivo? Or the rapid dissolving of drug from the nanocrystal into bile micelle is more important in vivo? The authors should discuss that issues in the revised manuscript.
Author Response
We carefully revised our manuscript according to the suggestions of the reviewer 2, and details are as follows.
< Q and A for Reviewer 2>
Q1. The authors should discuss a bit more detail about novelty of the work, as it seems that they have used a similar concept that was obtained in their previous works and that drug compound was just changed to IRB.
A1. Thank you very much for pointing this out. We have previously prepared nanocrystal dispersions by using similar treatment and additives. However, even with the same treatment, if the drug is different, the resulting particle size and absorption mechanism are different. Furthermore, the crystal structure is also different. Therefore, it is important to clarify the physical characteristics, absorption, and drug efficacy of nanocrystalline using the drugs with different properties for designing and application of nanocrystal dispersions. From this background, this study using irbesartan ARB, is classed as a BCS class II drug, will lead the way for further studies aiming to develop oral nanomedicines. In order to respond to the reviewer’s comment, we added the usefulness and importance in the Introduction (line 72-80).
Q2. In vitro penetration study. Did you also evaluate TEER in that chamber study? If not, a possibility to change the paracellular permeability in these treatments should be discussed in the manuscript.
A2. The reviewer’s comments are very important. The TEER was used to evaluate the membrane condition of culture cell. However, in the in vitro intestinal penetration study, we used isolated small intestines of rat instead of culture cells. Therefore, the TEER was not shown in this study. On the other hand, if membrane damage is caused, drug permeation will increase sharply (burst). In this study, the burst was not observed. The results suggested that the intestinal membrane was normal during the measurement period. We added the contents in the Materials and Methods (line 175-178).
Q3. Rat pharmacokinetic and pharmacodynamic studies. As the authors know, prandial states (fasted or fed) generally have a big impact on solubility of poorly soluble drugs in the gut, so these information should be clarified in Method.
A3. The reviewer’s comment is correct. The measurement of irbesartan concentration in rat blood was performed by using the rats fasted for 8 h. We added the content in the Materials and Methods (line 184).
Q4. Figure 4. A comparison between IRB-NC and IRB-P under 37 deg-C should also be performed.
A4. Thank you for pointing out this. The AUC0-6h of IRB-P was 32.9±6.7 nmol∙h/cm2 at 37 ºC. In order to respond to the reviewer’s comment, we compared the intestinal penetration between IRB-NC and IRB-P under 37 ºC. We added the data in the Results (line 274-276).
Q5. Figure 5. Rats generally have a high bile concentration in the small intestine even in fasted state. I guess IRB has a high log P value and would be easily incorporated into the bile micelles. Although according to the in vitro study CME have an important role in penetration of the drug in the intestinal epithelium, do the authors think that the CME also is a major factor in in vivo? Or the rapid dissolving of drug from the nanocrystal into bile micelle is more important in vivo? The authors should discuss that issues in the revised manuscript.
A5. The reviewer’s comment is correct. It was known that the dissolution rate in nanocrystalline suspensions was enhanced in comparison with drug microparticles. Therefore, the rapid dissolution of drugs from nanocrystal into bile micelle may be also related to high plasma irbesartan concentration in the rat administered IRB-NC. In order to respond to the reviewer’s comment, we added the possibility in the Discussion. Thank you very much for pointing this out (line 411-414)
Thank you for great comments.

Round 2
Reviewer 1 Report
The manuscript has been improved. It appears to be acceptable.
Reviewer 2 Report
I have no further comment.